# Whole Genome Development of Specific Alien-Chromosome Oligo (SAO) Markers for Wild Peanut Chromosomes Based on Chorus2

**DOI:** 10.3390/plants14193114

**Published:** 2025-10-09

**Authors:** Haojie Sun, Chunjiao Jiang, Weijie Qi, Yan Chen, Xinying Song, Chuantang Wang, Jing Yu, Guangdi Yuan

**Affiliations:** 1Shandong Peanut Research Institute (SPRI), Qingdao 266100, China; s994433@163.com (H.S.);; 2Peking University Institute of Advanced Agricultural Sciences, Shandong Laboratory for Advanced Agricultural Sciences at Weifang, Weifang 261325, China

**Keywords:** cultivated peanut, peanut wild species, specific alien-chromosome oligo (SAO) marker, Chorus2, alien-chromosome

## Abstract

The cultivated peanut (*Arachis hypogaea* L.) is a globally important oilseed and economic crop, but its narrow genetic base limits breeding progress. Wild *Arachis* species represent valuable genetic resources for enhancing the resilience of the peanut cultigen. While wild species from section *Arachis* are widely used in breeding programs, the detection of alien chromosomes in hybrids remains challenging due to limited molecular tools. In this study, a cost-effective and efficient system was established for generating species-specific molecular markers using low-coverage next-generation sequencing data, bypassing the need for whole-genome assembly. Utilizing the Chorus2 software, specific alien-chromosome oligo (SAO) markers were developed for four wild species, *A. duranensis* (accession A19), *A. pusilla* (A10), *A. appresipilla* (A33), and *A. glabrata* (G2 and G3). A total of 1166 primer pairs were designed, resulting in 220 SAO markers specific to *A. duranensis*, 77 to *A. pusilla*, 112 to *A. appresipilla*, 69 to *A. glabrata* G2, and 59 to *A. glabrata* G3, with the highest development efficiency observed in *A. duranensis* (55.0%). These markers span all chromosomes of the five wild accessions. Genome-wide, chromosome-specific SAO markers enable the efficient detection of introgressed alien chromosomes and provide insight into syntenic relationships among homoeologous chromosomes. These markers offer an effective tool for identifying favorable genes and facilitating targeted introgression for the genetic improvement of the cultivated peanut.

## 1. Introduction

The genus *Arachis* consists of 83 species and is divided into nine sections [1,2,3], encompassing 15 genomes including A, B, C, D, E, K, F, G, H, Ex, P, R_1_, R_2_, T, and Te [4,5,6,7]. Among these, the cultivated peanut (*Arachis hypogaea* L., 2*n* = 4x = 40, AABB) [4,5] serves as a globally important oilseed and economic crop. However, during domestication, it has undergone successive genetic bottlenecks, which, coupled with the heavy reliance on a narrow pool of core parental lines in breeding programs, have resulted in severe genetic erosion, leaving the peanut cultigen with limited genetic diversity and heightened vulnerability to diseases and pests [8].

Wild *Arachis* relatives of peanuts exhibit great diversity in agronomically important traits and have good adaptability to adverse environments. In particular, they possess desirable traits such as high oil content, high protein content, and high resistance to various biotic and abiotic stresses [9,10,11]. These traits make wild species invaluable genetic resources for expanding the genetic base of the cultivated peanut.

For the utilization of these elite genes, chromosome engineering has been used to produce alien chromosomes. Irrespective of the enormous genetic variation in wild germplasm and sophisticated techniques available for alien gene transfer [12,13], it is still difficult to efficiently identify introgressed chromatin when an alien chromosome was used in peanuts’ improvement. Although molecular markers offer a powerful solution, the current repertoire of species-specific markers for wild *Arachis* remains extremely limited. A few Transposable Element (TE) markers have been reported to distinguish *A. glabrata* from cultivated peanut [13,14], and similar Simple Sequence Repeat (SSR) markers have been used to identify chromosomes of wild species within the *Arachis* section [12]. Single Nucleotide Polymorphism (SNP) markers have also been employed to differentiate *A. stenosperma* from the cultivated peanut and to map the stem rot resistance gene in *A. stenosperma* [15]. It is far from enough to use these markers to identify the chromosomes of wild species, which seriously hinders the application of wild species genetic resources in peanut genetic improvement, creating an urgent need for high-throughput marker development.

In contrast, wheat and its relatives have seen the emergence of numerous markers specifically developed for wild species, which can serve as a reference for marker development in peanuts. Insertion Target (IT) markers, designed exclusively for wild germplasm, have already been successfully applied in *Dasypyrum villosum* [16,17]. Expressed Sequence Tag (EST) markers, which primarily target variations in the coding sequence (CDS), are also widely used for chromosome identification in both common wheat and *D. villosum* [18,19]. These advanced systems, however, depend on high-quality genome assemblies, a resource largely unavailable for most wild *Arachis* species. This limitation renders traditional marker development approaches costly, inefficient, and time-consuming for wild peanuts, while also producing an insufficient number of markers for comprehensive applications.

Therefore, based on the concept of abandoning the cumbersome procedures of sequencing and assembly required by traditional markers, this study tries to develop a new type of wild species-specific molecular SAO (specific alien-chromosome oligo) markers. Chorus2 can precisely help us with specific oligonucleotide calculations and screening.

Chorus2 is purpose-built software for identifying specific oligonucleotides [20]. Chorus2 has been proven to be highly effective in removing all duplicate elements in single-copy oligonucleotide screening. The tool can also retain oligos that are conserved across related species, thereby broadening their applicability within a phylogenetic clade. Consequently, it can be used to calculate single copy oligonucleotide sequences of wild species without the need for complete genome assembly. To date, however, Chorus2 has been employed almost exclusively for designing chromosome-painting probes, with libraries already developed for several species, including cucumber [21,22], banana [23], strawberry [24], potato [25,26], poplar [27], sugarcane [28,29], maize [30,31], citrus [32], and wheat [33,34]. Beyond probe development, the software holds considerable, yet largely untapped, potential for other areas of molecular cytogenetics.

In this study, we report a new method for developing chromosome-specific SAO markers for wild peanuts. This method identifies computationally specific sites using Chorus2 and efficiently develops a series of chromosome-specific SAO markers without sequence assembly, in order to utilize the beneficial genes from wild relatives for peanut breeding.

## 2. Results

### 2.1. Development of PCR-Based SAO Markers on Chromosomes A^du^01 and A^du^04 in A. duranensis

In this study, the feasibility of the method was first validated on chromosomes A^du^01 and A^du^04 in the *A. duranensis* accession A19. Chorus2 was used to screen the reference genome of the cultivated peanut Tifrunner for specific oligonucleotide short sequences. The pools of chromosome-specific oligonucleotide short sequences for A01 and A04, containing 85,185 and 81,930 sequences, were obtained, respectively. These sequences can not only be used to design probes for chromosome painting but also to develop specific markers for chromosome identification. Subsequently, the ChorusNoRef module within Chorus2 was employed to calculate chromosome-specific oligonucleotide short sequences for A^du^01 and A^du^04 in *A. duranensis* A19. Compared with the cultivated peanut Tifrunner, A19 exhibited 3551 Insertion-Deletions (InDels) and 48,384 SNPs on chromosome A01, and 3153 InDels and 45,459 SNPs on chromosome A04 (Appendix A). All these variants mapped unambiguously to the Tifrunner reference genome. Mapping the InDel variants to the reference genome revealed that they are concentrated mainly at the distal ends of chromosomes A^du^01 and A^du^04, whereas the pericentromeric regions show few differences (Table 1 and Figure 1A). To verify the performance of the SAO markers system, the markers were designed using the online software Primer 3 V0.4.0. A total of 104 evenly distributed InDel loci on each of chromosomes A^du^01 and A^du^04 were selected, primer pairs were designed, and the feasibility of SAO markers was validated (Figure 1B).

### 2.2. Validation and the Efficiency of SAO Markers Development by PCR

The 208 primer pairs (Appendix A) were used to amplify genomic DNA from *A. duranensis* accession A19 and the cultivated peanut HY665 by PCR (Figure 1D,E). Amplicons were detected in at least one accession for the vast majority of primer pairs; only 16 pairs (nine on chromosome A^du^01 and seven on chromosome A^du^04) produced no visible bands in either A19 or HY665. Subsequent analysis of the amplification products revealed clear polymorphisms between A19 and HY665 for 48 markers on chromosome A^du^01, examples include A19chr1-7, A19chr1-8, A19chr1-12, A19chr1-14, A19chr1-64, A19chr1-65, A19chr1-66, A19chr1-68, A19chr1-70, A19chr1-71, A19chr1-73, and A19chr1-74, etc. (Figure 1D), yielding a polymorphism rate of 46.2% (Table 1 and Appendix A). Similarly, 51 markers on chromosome A^du^04 exhibited distinct differences, such as A19chr4-31, A19chr4-32, A19chr4-33, A19chr4-34, A19chr4-37, A19chr4-38, A19chr4-40, A19chr4-41, A19chr4-44, A19chr4-45, A19chr4-47, A19chr4-49, A19chr4-50, A19chr4-51, and A19chr4-52, etc. (Figure 1E), corresponding to a polymorphism rate of 49.0% (Table 1 and Appendix A). When these SAO markers were remapped to chromosomes A^du^01 and A^du^04, they were found to be distributed relatively evenly along each chromosome, mirroring the uniform spacing of the original primer designs and showing no pronounced clustering in any particular region (Figure 1C).

### 2.3. Development of SAO Markers in Five Wild Peanut Species

We applied the SAO marker system to the other chromosomes of *A. duranensis* and to three other wild peanut species from different sections. For *A. duranensis* accession A19, we only calculated the InDel polymorphic sites of the A genome in cultivated peanut relative to its corresponding chromosomes, excluding chromosomes A^du^01 and A^du^04. A total of 25,872 InDel polymorphic sites were obtained for the remaining chromosomes. For the diploid wild peanut *A. pusilla* accession A10 and *A. appressipila* accession A33, we aligned their single chromosome sets to the A and B genomes of cultivated peanut to calculate polymorphic sites. For example, chromosome H01 of accession A10 was aligned to both chromosomes A01 and B01 of cultivated peanut, and similarly, chromosome P^R^01 of accession A33 was aligned to both chromosomes A01 and B01. In this way, *A. pusilla* accession A10 obtained a total of 32,392 InDel polymorphic sites, and *A. appressipila* accession A33 obtained a total of 44,495 InDel polymorphic sites. For the tetraploid wild peanut *A. glabrata* accessions G2 and G3, we aligned their two chromosome sets to the A and B genomes of cultivated peanut. For example, chromosome R_1_01 of accession G2 or G3 was aligned to chromosome A01 of cultivated peanut, and chromosome R_2_01 was aligned to chromosome B01. The *A. glabrata* accession G2 obtained a total of 1893 InDel polymorphic sites, and *A. glabrata* accession G3 obtained a total of 25,191 InDel polymorphic sites (Table 1 and Appendix A).

To specifically identify each chromosome of the five different wild peanut accessions, in this study, at least 150 primer pairs were designed for each wild peanut accession (Table 1 and Appendix A). Through the PCR amplification of 958 primer pairs, 438 specific markers were screened (Appendix A). Based on the banding patterns amplified by agarose gel electrophoresis, the SAO markers were classified into four different types: Type I, wild peanut accessions have electrophoretic bands, while cultivated peanut does not have electrophoretic bands; Type II, wild peanut accessions do not have electrophoretic bands, while cultivated peanut has electrophoretic bands; Type III, the electrophoretic bands of wild peanut species and cultivated peanuts are at different migration rates; Type IV, both wild peanut species and cultivated peanuts lack electrophoretic bands, or the bands are at the same migration rate (Figure 2A and Appendix A).

Markers of Type I, Type II, and Type III all exhibited polymorphism between wild and cultivated peanuts, while markers of Type IV showed no polymorphism between wild and cultivated peanuts. However, markers of Type II only produced PCR amplification bands in cultivated peanuts and could not specifically identify wild peanuts. For the precise tracking of chromosomes in wild peanut species, it is recommended to use Type I and a subset of Type III SAO markers.

### 2.4. The Efficiency of the SAO Markers’ Development by PCR in Five Wild Peanut Species

We screened specific amplicons corresponding to group 1 to group 10 for five different wild peanut accessions, A19, A10, A33, G2, and G3 (Figure 2B,C). These markers could make the identification of wild peanut chromosomes more convenient and accurate. For the design of SAO markers, there is a significant difference in success rates among the five wild species. A19 and A33 have higher success rates, at 55.0% and 52.1%, respectively, which are much higher than the three wild species A10, G2, and G3 (38.5%, 34.5%, and 39.1%) (Table 1).

### 2.5. Identification of the Chromosomal Composition of Wild Peanut and Cultivated Peanut Hybrid F_1_ Generation Using SAO Markers

In our previous studies, hybrids between incompatible wild Arachis species and cultivated peanuts were obtained through in situ embryo rescue. A cross between accession A33 of *A. appresipilla* and the cultivar HY665 produced three prostrate F_1_ hybrids: 24SHJ1-12, 24SHJ1-13, and 24SHJ1-14, while accession G2 of *A. glabrata* crossed with HY665 yielded two additional prostrate F_1_ plants, 24SHJ13-4 and 24SHJ14-10 (Figure 3A). To determine how many chromosomes from the wild species had been introgressed into the cultivated peanut genome, we selected ten A33 specific SAO markers, one from each homoeologous group (groups 1 to 10), for PCR analysis. All three F_1_ hybrids (24SHJ1-12, 24SHJ1-13, and 24SHJ1-14) produced the expected SAO markers amplicons for all ten homoeologous groups (Figure 3B). Likewise, the two F_1_ hybrids 24SHJ13-4 and 24SHJ14-10 yielded 20 G2 specific SAO markers amplicons corresponding to the ten homoeologous groups (Figure 3C).

## 3. Discussion

In this study, we leveraged Chorus2 to develop, without genome assembly, a new SAO marker system that specifically tracks chromosomes of wild *Arachis* species. Like SSR and IT markers, all three marker types must first identify polymorphic sites between the wild and cultivated genomes so that primer pairs can be designed, greatly increasing success rates. SSR markers were designed in the wild peanut species *A. duranensis* and obtained only 61 specific markers, a success rate of 20.3% [35]. Owing to the limitations inherent in wild peanut genomes, IT markers have not yet been applied to chromosome identification in *Arachis*; nevertheless, their success rate in *D. villosum* relative to common wheat can reach 51.79% [17]. The SAO-marker system can calculate primer-design sites without genome assembly, thereby circumventing the constraints imposed by wild-genome resources. After the amplification and screening of primer pairs, the polymorphism rate of SAO markers developed for *A. duranensis* reached 55.0%, markedly higher than that of SSR markers. The successful development of the SAO marker system in peanuts offers a low-cost, novel approach for developing wild species-specific molecular markers in other crops, such as wheat, rice, potato, and foxtail millet.

Specific SAO markers for four wild *Arachis* species (*A. duranensis*, *A. pusilla*, *A. appresipilla,* and *A. glabrata*) were successfully designed, but the number of polymorphic sites identified, and the resulting marker-design success rates varied considerably among them. This variation is closely related to the phylogenetic distance between each wild species and cultivated peanuts: the closer the relationship, the easier it is to identify informative polymorphic sites and the higher the marker-design success rate. A similar observation was reported during the development of the Chorus2 software; when using the ChorusNoRef pipeline to design wild-potato probes for chromosome painting, they found that the more distantly related the wild potato was to the cultivated one, the weaker the Fluorescence in situ Hybridization (FISH) signal became [20]. ChorusNoRef selects single-copy short sequences by alignment against a reference genome; thus, the more distant the target species is from the reference, the more difficult it is to map its sequences to that genome.

Comparative genomic resequencing analysis involving *A. duranensis* varieties showed that *A. duranensis* from Argentina is more closely related to the A subgenome of Tifrunner [36,37]. Previous reports had shown that through FISH karyotyping and distant hybridization, *A. duranensis* and *A. appresipilla* were more similar and closely related to cultivated peanuts [12]. Taxonomic analyses also placed *A. pusilla* in a more ancient lineage, indicating a more distant relationship with cultivated peanut [38,39,40]. Full-length transcriptome analysis further revealed a marked divergence between *A. glabrata* and cultivated peanut [41]. In the present study, the relatedness of four wild Arachis species to cultivated peanuts was assessed by quantifying divergent oligo sites and evaluating the success rates of the corresponding SAO marker designs. *A. duranensis* and *A. appresipilla* exhibited a relatively high number of divergent oligo sites, yet their SAO markers design success rates exceeded 50%, implying a closer genetic affinity to cultivated peanut. In contrast, *A. pusilla* and *A. glabrata* showed SAO marker success rates below 40%, indicating a more distant relationship. These findings align well with earlier reports.

Recent studies have demonstrated that analyzing sequence divergence with short oligonucleotide fragments is far more powerful than direct reads-to-reference alignment. Such fragments capture a far richer spectrum of genetic variation. *k*-mer-based GWAS [42,43], a new association strategy that indexes genomes as *k*-mers and performs association tests on these units, consistently outperformed conventional SNP-based GWAS. Following the same principle, the present study employed short oligo sequences to quantify divergence. After segmentation and computation with Chorus2 [44], analogous to *k*-mer decomposition, oligo sequences that are highly specific and capable of pinpointing indels were obtained, traits absent in raw sequencing reads. This specificity enabled SAO markers to be amplified with markedly higher fidelity.

Moreover, an intriguing pattern was observed: when the numbers of chromosome-specific oligos were tallied across the ten homeologous groups (groups 1 to groups 10), group 3 contained markedly more unique oligos than any other group (Appendix A). This finding warrants further investigation.

This study introduces the SAO marker system, a low-cost, high-throughput approach for identifying chromosomes in wild *Arachis* species. Genome-wide markers spanning all chromosomes of the wild relatives have been developed; they not only facilitate the detection of alien chromatin introgressed into the cultivated peanut background but also provide evidence of homeologous relationships between chromosomes. These chromosome-specific markers offer breeders an efficient tool for selecting beneficial genes located on individual chromosomes.

## 4. Materials and Methods

### 4.1. Plant Materials

The one cultivated peanut variety Huayu 665 (HY665), bred by the Shandong Peanut Research Institute and 5 wild *Arachis* species/accessions including *A. pusilla* (A10), *A. duranensis* (A19), *A. appresipilla* (A33), and *A. glabrata* (G2, G3) were used in this study. Their genome constitutions are listed in Table 1. The wild accessions with A- and G-prefixed numbers were kindly provided by the Guangxi and Guangdong Academies of Agricultural Sciences, respectively. Additionally, five interspecific F_1_ hybrids were also included in this study: three (24SHJ1-12, 24SHJ1-13, and 24SHJ1-14) derived from HY665 × A33 crosses, and two (24SHJ13-4 and 24SHJ14-10) from HY665 × G2 crosses (Table 2). All peanut materials were maintained at the Shandong Peanut Research Institute. These materials were used to analyze the chromosome specificity of the molecular markers.

### 4.2. DNA Extraction and Next-Generation Sequencing (NGS)

Genomic DNA was extracted from 0.5 g of fresh leaves using the CTAB method, followed by purification to remove RNA, amylase, and other unwanted components. Purified DNA samples were stored at −20 °C until further use.

The next-generation sequencing of wild peanuts was performed at a sequencing depth of 10×. In this study, 5 wild *Arachis* species/accessions (A10, A19, A33, G2, and G3) were sequenced separately. According to the protocol of Annoroad Universal EZ DNA Library Prep kit for MGI V1.0 (CAT: AN210105-S, Annoroad Gene Technology, Beijing, China) we used Bioruptor Pico to conduct DNA fragmentation treatment on the qualified DNA samples, then added Adapters for MGI. The fragment size was selected, and the DNA library was obtained by PCR enrichment. Finally, the cyclization of single stranded DNA was completed. The library was put into DNBSEQ-T7 for sequencing, and 150 bp double ended sequencing readings were obtained. The three wild peanut lines A10, A19, and A33 each yielded approximately 14 Gb of sequencing data, while the other two wild lines, G2 and G3, generated about 26 Gb each.

### 4.3. Specific Loci Calculation

Based on the A and B subgenome of cultivate peanuts on the Tifrunner genome database (https://data.legumeinfo.org/Arachis/hypogaea/genomes/Tifrunner.gnm2.J5K5/, (accessed on 30 August 2025)), the specific oligo sequences in Tifrunner were developed using Chorus2 (https://github.com/zhangtaolab/Chorus2, (accessed on 30 August 2025)). The A and B subgenome sequences were divided into oligos of 45 nt in a step size of 25 nt. Oligos were firstly filtered out if they were mapped to two or more locations with 80% homology. The *k*-mer method was further used to filter out repetitive sequences to ensure each oligo was a single copy. Oligos with dTm < 10 °C (dTm = Tm−hairpin Tm) were also further eliminated. This yielded subgenome-specific oligo pools for the A and B subgenomes of Tifrunner. ChorusNoRef uses short sequence reads, such as those from Illumina or MGI sequencing, which can be readily generated with a minimum cost. Subsequently, the ChorusNoRef module within the Chorus2 pipeline was employed to map these subgenome-specific oligo pools against the sequenced reads of the wild *Arachis* species. Short reads from the target species are first aligned to the reference genome. Reads that overlap reference-derived oligos are then used for local assembly, thereby retrieving the corresponding genomic segments from the target species. Finally, new oligos are designed by substituting the reference sequences with the target-specific sequences. This new oligo pools sequence contains sequence differences between the target species and Tifrunner (Figure 4).

### 4.4. Primer Design

Primers were designed on two segments of oligonucleotides that were calculated to be less than 500 nt apart. For each primer pair, the forward primer was positioned on the left oligonucleotide and the reverse primer on the right oligonucleotide, with the requirement that at least one primer of each pair must be located within a region showing sequence differences between wild and cultivated peanuts (Figure 4). A total of 1166 SAO primer pairs were designed using the online software Primer 3 V0.4.0 (http://frodo.wi.mit.edu/primer3/, (accessed on 10 November 2024)). All the primer pairs were synthesized by General Biol (Anhui) Co., Ltd., Chuzhou, China.

### 4.5. PCR Amplification and Products Visualization

PCR amplification was conducted in a 10 µL reaction containing 40 ng genomic DNA, 2 µM each of the primer pairs, and 5 µL 2 × Taq Master Mix (Vazyme, Nanjing, China, P111) with an OSE-GP-03 thermal cycler (TIANGEN, Beijing, China). The samples were denaturated at 94 °C for 5 min and subjected to 32 cycles of the following: 30 s of denaturation at 94 °C, 30 s at 52–60 °C, according to the different primers, and a 40 s elongation at 72 °C. A final cycle with an extension of 10 min at 72 °C was applied to complete the reactions. The PCR products were separated in 2% agarose gels and visualized with Red nucleic acid dye.

## Figures and Tables

**Figure 1 plants-14-03114-f001:**
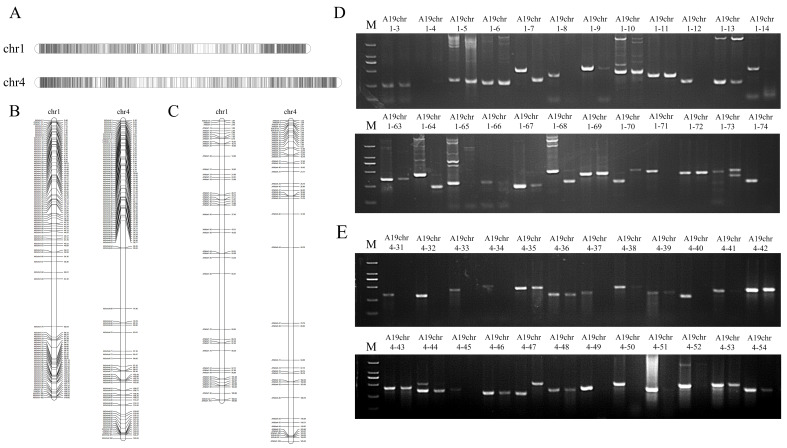
Differential oligonucleotide-locus distribution on chromosomes A^du^01 and A^du^04 of *A. duranensis* and the design, amplification of SAO markers. (**A**): Distribution of differential oligo loci on chromosomes chr1 and chr4 in *A. duranensis*. (**B**): Distribution of the 208 designed primer pairs on chromosomes chr1 and chr4. Marker names are shown on the (**left**) and their chromosomal physical positions (Mb) on the (**right**). (**C**): Distribution of SAO markers showing amplification differences between *A. duranensis* and cultivated peanut HY665 on chromosomes A^du^01 and A^du^04. Marker names are on the (**left**) and their physical positions (Mb) on the (**right**). (**D**): Electrophoretic profiles of 24 SAO markers on chromosome A^du^01. M indicates the DL2000 DNA marker. The left lane of each pair shows the PCR amplification band in *A. duranensis* DNA; the right lane shows the band in HY665 DNA. (**E**): Electrophoretic profiles of 24 SAO markers on chromosome A^du^04. M indicates the DL2000 DNA marker. The left lane of each pair shows the PCR amplification band in *A. duranensis* DNA; the right lane shows the band in HY665 DNA.

**Figure 2 plants-14-03114-f002:**
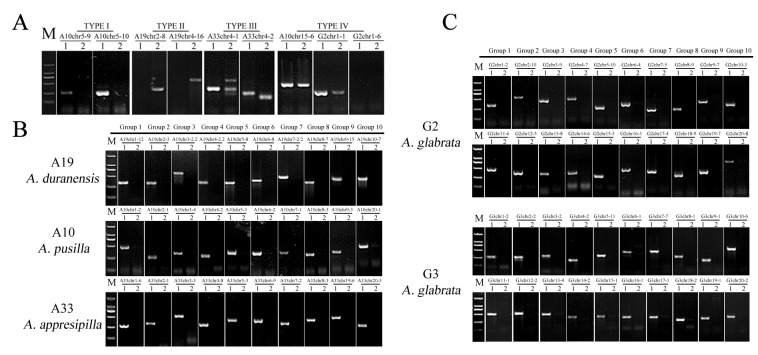
Four types of SAO markers and the specific SAO markers bands for five wild *Arachis* species. (**A**): Four types of SAO markers. (**B**): Specific SAO marker bands for the ten chromosomes of *A. duranensis*, *A. pusilla*, and *A. appresipila*. (**C**): Specific SAO marker bands for the twenty chromosomes of the two *A. glabrata* accessions, G2 and G3. M: DL2000 DNA marker. 1: wild *Arachis* species. 2: HY665.

**Figure 3 plants-14-03114-f003:**
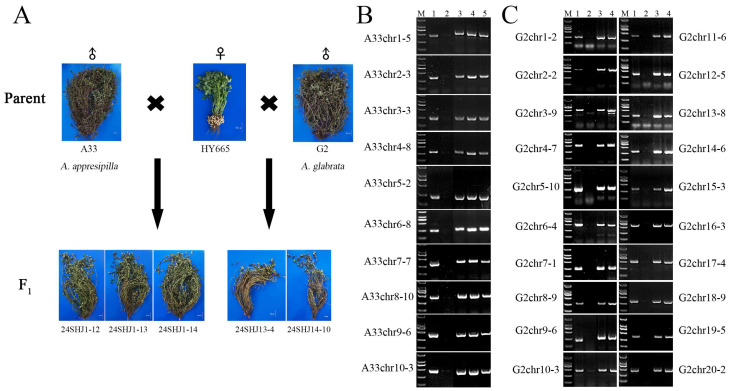
Progeny plants from the cross of HY665 × A33, HY665 × G2, and their SAO markers analyzed. (**A**): Photographs of progeny from the crosses HY665 × A33 and HY665 × G2, along with the parental plants A33, G2, and HY665, bar = 10 cm, (**B**): Amplification bands of SAO markers in A33, HY665, and their F_1_ progeny (HY665 × A33), M: DL2000 DNA marker, 1: A33. 2: HY665, 3: 24SHJ1-12, 4: 24SHJ1-13, 5: 24SHJ1-14. (**C**): Amplification bands of SAO markers in G2, HY665, and their F_1_ progeny (HY665 × G2), M: DL2000 DNA marker, 1: G2, 2: HY665, 3: 24SHJ13-4, 4: 24SHJ14-10.

**Figure 4 plants-14-03114-f004:**
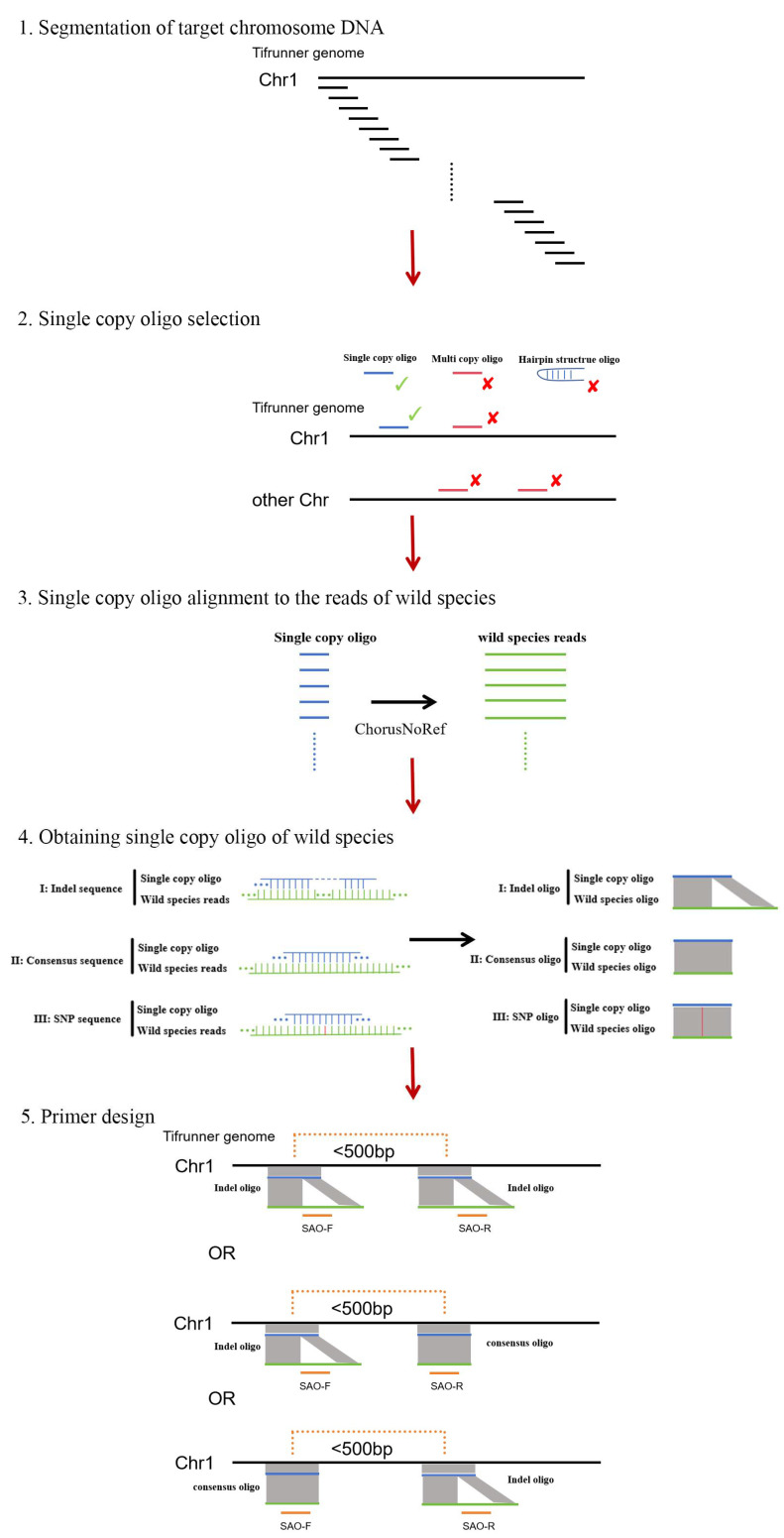
Schematic diagram of SAO marking system principle.

**Table 1 plants-14-03114-t001:** Differential loci and polymorphic markers between wild and cultivated peanuts.

Peanut Wild Species/Chromosome	Number of Indel Positions	Number of Design Markers	Number of Specific Markers	Marker Polymorphism Rate
A19 chromosomeA^du^01	3551	104	48	46.2%
A19 chromosomeA^du^04	3153	104	51	49.0%
A19 (Excluding chromosomeA^du^01 and A^du^04)	25,872	192	121	63.0%
A19 (total)	32,576	400	220	55.0%
A10	32,392	200	77	38.5%
A33	44,495	215	112	52.1%
G2	1893	200	69	34.5%
G3	25,191	151	59	39.1%

**Table 2 plants-14-03114-t002:** Peanut materials used in this study and their identification numbers.

Material No.	Chromosomal Constitutions	Species	Section
HY665	2*n* = 4x = 40, AABB	*A. hypogaea*	*Arachis*
A10	2*n* = 2x = 20, HH	*A. pusilla*	*Heteranthae*
A19	2*n* = 2x = 20, AA	*A. duranensis*	*Arachis*
A33	2*n* = 2x = 20, P^R^P^R^	*A. appresipilla*	*Procumbentes*
G2	2*n* = 4x = 40, R_1_R_1_R_2_R_2_	*A. glabrata*	*Rhizomatosae*
G3	2*n* = 4x = 40, R_1_R_1_R_2_R_2_	*A. glabrata*	*Rhizomatosae*
24SHJ1-12	*n* = 3x = 30, ABP^R^	*A. hypogaea*-*A. appresipilla*	-
24SHJ1-13	*n* = 3x = 30, ABP^R^	*A. hypogaea*-*A. appresipilla*	-
24SHJ1-14	*n* = 3x = 30, ABP^R^	*A. hypogaea*-*A. appresipilla*	-
24SHJ13-4	*n* = 4x = 40, ABR_1_R_2_	*A. hypogaea*-*A. glabrata*	-
24SHJ14-10	*n* = 4x = 40, ABR_1_R_2_	*A. hypogaea*-*A. glabrata*	-

## Data Availability

Data derived from public domain resources: the data presented in this study are available at [https://data.legumeinfo.org/Arachis/hypogaea/genomes/Tifrunner.gnm2.J5K5/, accessed on 4 October 2025], reference number [Tifrunner.gnm2.J5K5]. The data obtained from this study’s analysis: data is contained within the article or Appendix A. The original contributions presented in this study are included in the article/Appendix A. Further inquiries can be directed to the corresponding author(s).

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
