# Peer review of "Whole Genome Development of Specific Alien-Chromosome Oligo (SAO) Markers for Wild Peanut Chromosomes Based on Chorus2"

_plants, 2025, doi:10.3390/plants14193114_

Round 1
Reviewer 1 Report
Comments and Suggestions for Authors
The manuscript entitled “Whole Genome Development of Specific Alien-Chromosome Oligo (SAO) Markers for Wild Peanut Chromosomes Based on Chorus2” presents a valuable study aimed at providing an effective tool for identifying favorable genes and facilitating targeted introgression for the genetic improvement of cultivated peanuts. The Specific Alien-Chromosome Oligo (SAO) Markers not only enable the detection of alien chromatin introgressed into the cultivated peanut background but also offer evidence of homeologous relationships between chromosomes.
Formal comments on the text:
Line 83: Remove the period before “To date”
Line 96: The authors should avoid using “our research team”; a more detailed unit should be specified.
Author Response
Thank you for your letter together with the comments and suggestions from the reviewers on our manuscript. We have carefully considered all the advices and revised the manuscript accordingly. Our responses or explanations are listed as followings. All the changes were highlighted using the "Track Changes" tool of MS Word in the revised manuscript. We hope we have addressed all the raised issues and got every point clearer. If any more questions, please contact us.
1. Line 83: Remove the period before “To date”
Reply: We have removed the period before “To date” in line 84
2. Line 96: The authors should avoid using “our research team”; a more detailed unit should be specified.
Reply: We have changed "our research team" to "Shandong Peanut Research Institute." in line 97-98.
Reviewer 2 Report
Comments and Suggestions for Authors
The article is very interesting, and I believe it represents a significant contribution for the peanut research community and/or for breeding proposes. It is well-written and presents the essays and results in a logical order.
Only a few very minor issues were identified, and comments on them were included directly in the original file.

Author Response
Thank you for your letter together with the comments and suggestions from the reviewers on our manuscript. We have carefully considered all the advices and revised the manuscript accordingly. Our responses or explanations are listed as followings. All the changes were highlighted using the "Track Changes" tool of MS Word in the revised manuscript. We hope we have addressed all the raised issues and got every point clearer. If any more questions, please contact us.
Only a few very minor issues were identified, and comments on them were included directly in the original file.
Reply: Thank you very much for your careful revision of the entire manuscript. We have addressed all the issues you raised. Regarding the selection of chromosomes chr1 and chr4, these two chromosomes were chosen at random to evaluate the feasibility of SAO marker development.
Reviewer 3 Report
Comments and Suggestions for Authors
This manuscript describes an important study on peanut. This is well written. However, it can be further improved by adding following information.
- Clearly mention where the plant materials were obtained from
- Please provide brief introduction the Chorus2 software
- What is the size/s of the Arachis hypogaea genome and how did you obtain 10x coverage, what was the output of genome sequencing? how much data generated?
- In the discussion section, one of the arguments was that ability of using this marker technique in identifying important genes. in your study have you identified any genes? it was clear enough.
- Reference can be added in line 37.
- Scientific names should be in italic
- Punctuations needed to be checked.
- is it possible to conduct QTL analysis on these two chromosomes to validate one or several markers identified in the study.
Author Response
Thank you for your letter together with the comments and suggestions from the reviewers on our manuscript. We have carefully considered all the advices and revised the manuscript accordingly. Our responses or explanations are listed as followings. All the changes were highlighted using the "Track Changes" tool of MS Word in the revised manuscript. We hope we have addressed all the raised issues and got every point clearer. If any more questions, please contact us.
- Clearly mention where the plant materials were obtained from.
Reply: We have already indicated the origins of all plant materials in line 97-104.
- Please provide brief introduction the Chorus2 software.
Reply: We describe the Chorus2 software in lines 79–89 of the manuscript.
- What is the size/s of the Arachis hypogaea genome and how did you obtain 10x coverage, what was the output of genome sequencing? how much data generated?
Reply: In this study, none of the wild Arachis accessions possessed publicly available genome information except A. duranensis. We therefore generated new MGI next-generation sequencing data (150-bp paired-end reads) for all materials. Genome-size estimation indicated that the diploid accessions A10, A19 and A33 are ~1.3 Gb, whereas the tetraploid accessions G2 and G3 are ~2.5 Gb. Total sequencing output reached ~14 Gb for the three diploids and ~26 Gb for the two tetraploids. These details are provided in lines 112–122 of the manuscript.
- In the discussion section, one of the arguments was that ability of using this marker technique in identifying important genes. in your study have you identified any genes? it was clear enough.
Reply: I'm very sorry, but I don't quite understand this issue; our manuscript does not claim that these markers were used for gene identification. I tried to explain this issue. In the Discussion (lines 334–343) we only cite another group’s K-mer-based GWAS study to illustrate that short-sequence markers can be developed more efficiently and accurately. A large proportion of the loci we targeted do fall within genic regions, so the markers may possess potential for gene tagging, but this was not demonstrated here. We appreciate your insightful comment and will explore the use of these markers for gene mapping and identification in future work.
- Reference can be added in line 37.
Reply: We have added the references [4, 5] in line 37.
- Scientific names should be in italic.
Reply: We have reviewed and revised the scientific names throughout the manuscript to ensure they are all in italics.
- Punctuations needed to be checked.
Reply: We have checked and corrected all punctuation marks in the manuscript to ensure their accuracy.
- Is it possible to conduct QTL analysis on these two chromosomes to validate one or several markers identified in the study.
Reply: I'm very sorry, but I don't quite understand this issue either,as no mention of QTL mapping appears in our manuscript. Here is the clarification we can offer: The wild Arachis species used here carry chromosome sets that are only partly homologous to those of cultivated peanut. After hybridization, pairing and recombination between the two chromosome sets are largely suppressed, so a conventional genetic map cannot be constructed. Physical mapping of the markers will therefore require the creation and screening of numerous chromosome-structure variants through cytogenetic approaches—work that lies beyond the scope of the present study. Fortunately, comparative genomics and homology-based analyses already allow us to assign the markers to approximate chromosomal positions. We appreciate your insightful suggestion and will pursue physical mapping in future research.